# Compact Neural Networks based on the Multiscale Entanglement Renormalization Ansatz

**Andrew Hallam[1], Edward Grant[2], Vid Stojevic[1,3], Simone Severini[2,4], Andrew G. Green[5]**

[1] *Department of Physics & Astronomy, University College London*
[2] *Department of Computer Science, University College London*
[3] *GTN Ltd.*
[4] *Institute of Natural Sciences, Shanghai Jiao Tong University*
[5] *London Centre for Nanotechnology, University College London*

## Abstract

This paper demonstrates a method for tensorizing neural networks based upon an efficient way of approximating scale invariant quantum states, the Multi-scale Entanglement Renormalization Ansatz (MERA). We employ MERA as a replacement for the fully connected layers in a convolutional neural network and test this implementation on the CIFAR-10 dataset. The proposed method outperforms factorization using tensor trains, providing greater compression for the same level of accuracy and greater accuracy for the same level of compression. We demonstrate MERA layers with 3900 times fewer parameters and a reduction in accuracy of less than $1\%$ compared to the equivalent fully connected layers, scaling like $\mathcal{O}(N^{\log_2 3})$.

## 1 Introduction

The *curse of dimensionality* is a major bottleneck in machine learning, stemming from the exponential growth of variables with the number of modes in a data set (Cichocki et al. (2016)). Typically state-of-the-art convolutional neural networks have millions or billions of parameters. However, previous work has demonstrated that representations stored in the network parameters can be highly compressed without significant reduction in network performance (Novikov et al. (2015), Garipov et al. (2016), Hinton et al. (2015)). Determining the best network architecture for a given task remains an open problem.

Descriptions of quantum mechanical systems raise a similar challenge; representing $n$ $d$-dimensional particles requires a rank-$n$ tensor whose memory cost scales as $d^n$. Indeed, it was the promise of harnessing this that led Richard Feynman (Feynman (1982)) to suggest the possibility of quantum computation. In the absence of a quantum computer, however, one must use compressed representations of quantum states.

A level of compression can be achieved by factorizing the tensorial description of the quantum wavefunction. Many such factorizations are possible, the optimal structure of the factorization being determined by the structure of correlations in the quantum system being studied. A revolution in quantum mechanics was made by realizing that the best way to characterize the distribution of correlations and information in a state is by a quantity known as *entanglement* – loosely the mutual quantum information between partitions of a quantum system (Eisert et al. (2010)).

This has led to many successful applications of tensorial approaches to problems in solid state physics and quantum chemistry over the past 25 years (Orús (2014), Kin-Lic Chan et al. (2007)). Intriguing ideas have also emerged over the past few years attempting to bridge the successes of neural networks in machine learning with those of tensorial methods in quantum physics, both at a fundamental level (Lin et al. (2017), Mehta & Schwab (2014)), and as a practical tool for network design (Levine et al. (2017)). Recent work has suggested that entanglement itself is a useful quantifier of the performance of neural networks (Levine et al. (2017), Liu et al. (2017))

The simplest factorization employed in quantum systems is known as the *matrix product state* (Orús (2014)). In essence, it expresses the locality of information in certain quantum states. It has already been adopted to replace expensive linear layers in neural networks – in which context it has been independently termed *tensor trains* (Oseledets (2011)). This led to substantial compression of neural networks with only a small reduction in the accuracy (Novikov et al. (2015), Garipov et al. (2016)).

Here we use a different tensor factorization – known as the *Multi-scale Entanglement Renormalization Ansatz* (MERA) – that encodes information in a hierarchical manner (Vidal (2008)). MERA works through a process of coarse graining or renormalization. There have been a number of papers looking at the relationship between renormalization and deep learning. MERA is a concrete realization of such a renormalization procedure (Vidal (2009)) and so possesses a multi-scale structure that one might anticipate in complex data. A number of works have utilized tree tensor network models that possess a similar hierarchical structure. However, they do not include the *disentangler* tensors that are essential if each layer of the MERA is to capture correlations on different length scales (Liu et al. (2017)).

In this work we employ MERA as a replacement for linear layers in a neural network used to classify the CIFAR-10 dataset. Our results show that this performs better than the tensor train decomposition of the same linear layer, and gives better accuracy for the same level of compression and better compression for the same level of accuracy. In Section 2 we introduce factorizations of fully connected linear layers, starting with the tensor train factorization followed by a tree-like factorization and finally the MERA factorization. In Section 3 we discuss how this is employed as a replacement for a fully connected linear layer in deep learning networks. Section 4 gives our main results and we note connections with the existing literature in Section 5. Finally, in Section 6 we discuss some potential developments of the work.

## 2 TENSOR FACTORIZATION OF LINEAR LAYERS

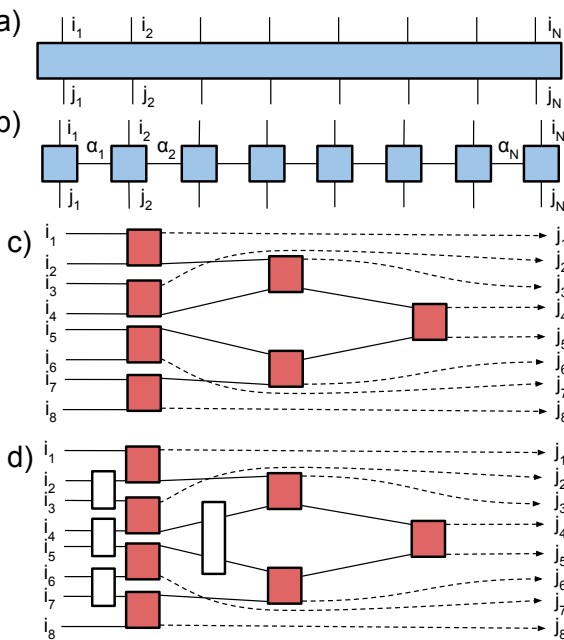

Figure 1: Schematic diagrams of various tensor factorizations of linear layers. a) a general linear layer, b) its tensor train factorization. The squares represent smaller tensors. Connections represent contractions as indicated in Eq.(1). c) Tree network factorization. d) MERA factorization.

In this report we have replaced the linear layers of the standard neural network with tensorial MERA layers. The first step in achieving this involves expressing a linear layer as a tensor. This can be accomplished by taking a matrix $\mathcal{W}$ and reshaping it to be a higher dimensional array. For

example, suppose $\mathcal{W}$ is $d^n$ by $d^n$ dimensional with components $\mathcal{W}_{AB}$. It can be transformed into a rank $2n$ tensor by mapping $A$ to $n$ elements $A \rightarrow i_1, i_2, ..., i_n$ and $B$ to another $n$ elements $B \rightarrow j_1, j_2, ..., j_n$. In this case each of the elements of the new tensor will be of size $d$.

Figure 1a gives a graphical representation of this rank $2n$ tensor $\mathcal{W}^{i_1,i_2,...,i_n}_{j_1,j_2,...,j_n}$. It is important to note that in this representation, the lines represent the indices of the tensors rather than weights. Figure 1b illustrates the tensor train decomposition of $\mathcal{W}$. This consists of writing the larger tensor as the contraction of a *train* of smaller tensors:

$$\mathcal{W}^{i_1,i_2,...,i_n}_{j_1,j_2,...,j_n} = \sum_{\alpha_1,\alpha_2,...,\alpha_{n-1}} A^{i_1}_{j_1,\alpha_1} A^{\alpha_1,i_1}_{j_1,\alpha_2} \cdots A^{\alpha_{n-1},i_n}_{j_n}. \tag{1}$$

In the tensor graphical notation, closed legs represent indices being summed over and free legs represent indices that aren't being summed over. For example, in equation 1 the $\alpha_i$ indices are being summed over and in Figure 1b the $\alpha_i$ lines are connected to tensors at both ends.

If each index runs over values from 1 to $d$, this represents an exponential reduction from $d^{2n}$ parameters to $n(Dd)^2$, where the indices $\alpha$ run over values from 1 to $D$ (known as the *bond order* or *Schmidt rank* in the quantum context). As noted above, this type of tensor factorization works well in physics when the information has a local structure (Eisert et al. (2010), Verstraete & Cirac (2006)); tensor trains capture correlations effectively up to length scales of order $\log D$ (Schollwöck (2011)). This means that while useful for many tasks, the learned representations will be highly local. Tensors at either end of a tensor train decomposition of a linear layer will not be strongly correlated with one another.

A hierarchically structured tensor network can better represent correlations across the linear layer. The tree tensor network shown in Figure 1c represents one possible hierarchical factorization. Each element of this network is a rank 4 tensor. The two tensors on the top left would have the form $\mathcal{M}^{j_1,\alpha_1}_{i_1,i_2}$ and $\mathcal{N}^{j_2,\alpha_2}_{i_3,i_4}$. The $i_n$ elements being represented by the lines on the left of the figure, the $j_n$ elements represented by the dotted lines on the right of the figure and the $\alpha_n$ lines being those connected with the tensor immediately to the right of $\mathcal{M}$ and $\mathcal{N}$.

Reading from left to right Figure 1c can be interpreted as follows: the tree-like connectivity imbues the network with a causal structure whereby a given linear element and its outputs are influenced by inputs in a region determined by its height in the tree.

For example, the rightmost element in Figure 1c is influenced by all of the inputs, whereas the top element in the middle column is influenced by inputs $i_1$ to $i_4$. Elements other than the rightmost tensor have one dashed output (that connects directly to the overall output) and one solid output (that ties it to the branching tree structure). These dashed lines are controlled by representations occurring on a particular scale in the data.

Notice that removing these dashed lines, the network has a true tree structure and represents a coarse graining or renormalization of the network. In this case, the linear elements are the *isometries* of the original MERA's definition (Vidal (2008; 2009)).

The simple tree network, which has been studied before in the context of neural networks, has a major deficiency. At each branching, it partitions the system in two, so that *in extremis*, the correlations between neighbouring inputs – for example $i_4$ and $i_5$ in Figure 1c – are only controlled by the element at the end of the network. Requiring the higher elements in the tree-structure to capture correlations between neighbouring inputs restricts their ability to describe the longer length scale correlations you would hope to capture by using a hierarchical structure.

The MERA (Vidal (2009)) factorization was introduced in order to solve this problem. As can be seen in Figure 1d it adds an additional set of rank 4 tensors called *disentanglers*. The MERA is constructed by taking a tree network and placing one of these rank 4 tensors $\mathcal{D}^{\beta_1,\beta_2}_{\gamma_1,\gamma_2}$ such that its right-going legs $\beta_1$ and $\beta_2$ connect to two adjacent tensors of the tree network. For example, if we consider the top left-most disentangler in Figure 1d it has elements $\mathcal{D}^{\beta_1,\beta_2}_{i_2,i_3}$ and connects to the tree elements $\mathcal{M}'^{j_1,\alpha_1}_{i_1,\beta_1}$ and $\mathcal{N}'^{j_2,\alpha_2}_{\beta_2,i_4}$ with $\beta_1$ and $\beta_2$ then being summed over.

The role of the disentanglers is to cause all correlations on the same length scale to be treated similarly. For example, correlations between any two neighbouring input indices $i_n$ and $i_{n+1}$ will be captured by either the first row of tree elements or the disentanglers. This allows the later elements in the network to work at capturing longer range correlations.

In summary, a rank-$N$ MERA layer can be constructed in the following manner:

1. Create a tree tensor layer. For example, an $N = 2^\tau$ tree can be constructed from $2^{\tau-1}$ rank-4 tree tensors $\mathcal{M}_{\gamma_1,\gamma_2}^{\beta_1,\beta_2}$ in the first layer, followed by $2^{\tau-2}$ tree tensors in the second layer until after $\tau$ layers there is only a single tree tensor.

2. A set of disentanglers are introduced. These are rank-4 tensors $\mathcal{D}_{\gamma_1,\gamma_2}^{\beta_1,\beta_2}$ which are placed such that every disentangler is contracted with two neighbouring tree tensors in an upcoming layer of the tree tensor.

## 3  EXPERIMENTS & NETWORK STRUCTURE

We have considered the performance of a neural network with the two penultimate fully connected layers of the model replaced with MERA layers, similar to the Novikov et al. (2015) study of compression of fully connected layers using tensor trains. We have quantified the performance of the MERA layer through comparisons with two other classes of networks: fully connected layers with varying numbers of nodes and tensor train layers with varying internal dimension. The three types of network are otherwise identical.

The networks consisted of three sets of two convolutional layers each followed by max pooling layers with $3 \times 3$ kernels and stride 2. The convolutional kernels were $3 \times 3$. There were 64 channels in all of the convolutional layers except for the input, which had three channels, and the last convolutional layer, which had 256 channels. The final convolutional layer was followed by two more hidden layers, these were either fully connected, MERA layers or TT-layers depending upon the network. The first of these layers was of size $4096 \times x$, the second is of size $x \times 64$. For all MERA and TT networks, these layers were $4096 \times 4096$ and $4096 \times 64$.

The final layer had 10 nodes corresponding to the 10 image classes in CIFAR-10. Leaky rectified linear units (LReLU) were used on all layers except the final layer, with $leak = 0.2$ (Maas et al. (2013)).

During training, nodes in the final convolutional layer and the two first fully connected layers were dropped with probability 0.5. The penultimate convolutional layer nodes were dropped with probability 0.2 (Srivastava et al. (2014)). Batch-normalization was used on all layers after dropout and max pooling (Ioffe & Szegedy (2015)). We did not use bias units.

Gaussian weight initialization was employed in the fully connected models with standard deviation equal to $\frac{1}{\sqrt{n_{in}}}$, where $n_{in}$ was the number of inputs (He et al. (2015)).

In this report we considered networks with two varieties of fully-connected layers. The first of these networks had a $4096 \times 4096$ fully connected layer followed by one which was $4096 \times 64$; this network was used as a benchmark against which the other models could be compared. The second network instead had a $4096 \times 10$ fully connected layer followed by a $10 \times 64$ layer. We trained this network to compare the MERA and tensor train layers to a fully connected model with a comparable number of parameters, in order to evaluate how detrimental naive compression is to accuracy.

A schematic of the two MERA layers can be found in Figure 2. The input to the first MERA layer was reshaped in to a rank-12 tensor with each index being dimension 2, as described in Section 2. The MERA layer was then constructed from a set of rank-4 tensors using the method described in Section 2.

The first MERA layer works as follows: It contains a column of 6 rank-4 tree elements, followed by 3 tree elements and finally a single tree element. 5 disentanglers are placed before the first column of tree elements and 2 more disentanglers are placed before the second column of tree elements.

The second MERA layer has an identical structure to the first MERA layer, one of the outputs of the first set of tree elements is fixed. As a result the output of the second MERA layer is 64 nodes. The dimensions of the internal indices in the MERA layers were allowed to vary, with a MERA-$D$ being a MERA layer with internal dimension $D$.

MERA weights were initialized using elements of randomized orthogonal matrices (Saxe et al. (2013)). The tensors themselves were constructed by reshaping these matrices, as described in Section 2. The random orthogonal matrix was constructed using the method of Stewart (Stewart

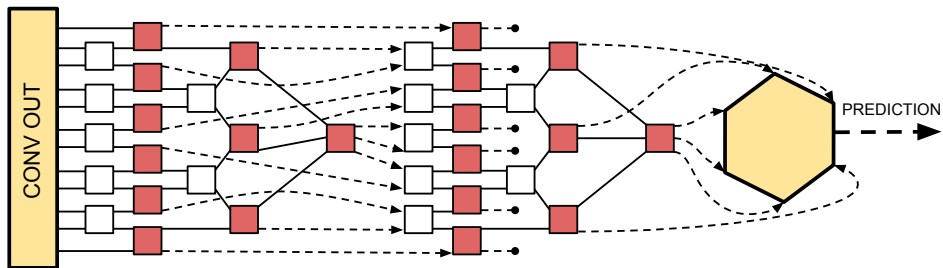

Figure 2: A schematic of the MERA layers of the model. The small rectangles represent linear elements to factorize a general linear layer. White rectangles represent disentanglers. Red rectangles represent tree elements. Solid black lines connecting nodes represent tensor contraction and dashed lines with arrow heads represent the nonlinearities being applied. Dashed lines ending in a circle represent fixed outputs.

(1980), Mezzadri (2007)). Starting from a random $n - 1 \times n - 1$ dimensional orthogonal matrix, a random $n \times n$ dimensional orthogonal matrix can be constructed by taking a randomly distributed $n$-dimensional vector, constructing its Householder transformation, and then applying the $n - 1$ dimensional matrix to this vector.

Finally, a network with its fully connected layers replaced with a tensor train decomposition was trained in order to provide a comparison with the MERA layers. The tensor train layers were constructed as described in Section 2 with the internal dimension being allowed to vary from $D = 3$ to $D = 5$. In the second tensor train layer, half of the output indices were fixed to match the second MERA layer.

We tested performance on the CIFAR-10 dataset. We used $50,000$ images for training and $10,000$ for testing. Each training batch consisted of $50$ images. Training data was augmented by randomly flipping and translating the input images by up to 4 pixels. Translated images were padded with zeros. All images were normalized by dividing by $255$ and subtracting the mean pixels value from the training set.

Test accuracy was recorded every $500$ iterations and training was stopped when test accuracy did not improve for 10 successive tests. The network was trained using backpropagation and the Adam optimizer, with initial learning rate $0.001$ (Kingma & Ba (2014)) and a softmax-cross-entropy objective.

The networks were implemented in Tensorflow r1.3 and trained on NVIDIA Titan Xp and 1080ti GPUs.

## 4 EXPERIMENTAL RESULTS

In Table 1 we compare the number of parameters used, the compression rate and the accuracy obtained for each of the models described in Section 3. The compression rate stated is with respect to the number of parameters used in the fully-connected benchmark model, FC-1.

When comparing the MERA networks to the fully connected model we can see a considerable drop in the number of parameters required with only a modest drop in the accuracy. MERA-2 compressed the fully connected layers by a factor of nearly 5500 with an accuracy drop of $1.48\%$. Increasing the internal dimension of the MERA to 3 we found the compression rate decreased to approximately 3900 but the accuracy was $86.2\%$, a drop of less than $1\%$ compared to the fully connected model. The compression rate of an entire MERA network compared to the fully connected network was approximately 57. We did not attempt to compress the convolutional layers in this work, and therefore the vast majority of parameters were focused in these layers for the MERA networks.

How significant is the MERA network structure we have chosen to the results obtained? The results of the MERA networks can be compared against the fully connected layer with only 10 nodes, see FC-2 in Table 1. This network possesses 100 times more parameters than MERA-2 but the drop in the accuracy is 6.04%, significantly more than any MERA network.

The MERA network also compares favourably to tensor train methods. TT-3 had a comparable compression rate to the MERA methods but a more significant drop in accuracy, 2.22%. For a tensor train network to achieve an accuracy drop of only 1% the internal dimension had to be increased to 7, resulting in an inferior compression rate compared to MERA-2.

In addition to the degree of compression achieved by these networks, we also address the time to optimize. There is evidently a degree of compromise required here: the number of multiplications required to apply a MERA layer scales with the input size $N$ and bond order $D$ as $N^{\log_2 D}$. The equivalent scaling for a tensor train and fully connected layer are $ND^2$ and $N^2$, respectively. This is reflected in the times taken to optimize these networks. Note however, that MERA can accommodate correlations at all scales of its input even at low bond order, whereas tensor trains require a bond order that scales exponentially with the length scale of correlation (Orús (2014)). MERA is, therefore, expected to scale better for very large data sets than either tensor trains or fully connected layers.

Table 1: The experimental results for the different models. FC1 was the fully-connected model and FC2 was the fully-connected model with severely reduced number of parameters in the fully-connected layers. MERA-2 and MERA-3 were MERA models with the MERA layers of internal bond dimension 2 and 3 respectively. Finally TT-3, TT-5, TT-7 were tensor train models with the internal dimension being 3, 5 and 7.

| Network | Parameters (Fully connected) | Parameters (Total) | Compression (Fully connected) | Compression (Total) | Accuracy |
|---------|------------------------------|--------------------|-------------------------------|---------------------|----------|
| FC-1 | 17041920 | 17338560 | 1 | 1 | 0.8719 |
| FC-2 | 305664 | 602304 | 55 | 28.79 | 0.8114 |
| MERA-2 | 3112 | 299752 | 5476 | 57.85 | 0.8579 |
| MERA-3 | 4342 | 300982 | 3924 | 57.61 | 0.8620 |
| TT-3 | 3168 | 299854 | 5272 | 57.82 | 0.8497 |
| TT-5 | 4380 | 301020 | 3891 | 57.60 | 0.8532 |
| TT-7 | 6088 | 302728 | 2799 | 57.27 | 0.8629 |

## 4.1 STUDY OF ABLATED NETWORKS

As can be seen in Table 1, the number of parameters in the fully convolutional layers far outweigh the parameters in the fully connected, MERA or tensor train layers of the network. In order to study the behaviour of the MERA and tensor train layers without the convolutional layers distorting the results we have created an ablated network with the number of parameters in the convolutional layers minimized.

In this case the only convolutional layer was had a $1 \times 1$ kernel with 3 channels in and 4 channels out. This was followed by $4096 \times 64$ MERA or tensor train layer. The final layer was a $64 \times 10$ layer as in the previous network structure. A MERA-2 layer was used, with 232 parameters. The tensor-train layer used a mixture of TT-2 and TT-3 so it would contain 236 parameters, as close as possible to the MERA-2 layer.

For this simplified network we found that the MERA-2 network achieved an accuracy of 47.36% ($\sigma = 0.48\%$) while the tensor train achieved an accuracy of 46.65% ($\sigma = 0.25\%$). Evidence of a MERA layer once again outperforming a tensor train layer with a comparable number of parameters.

## 5 RELATED WORK

Given how memory intensive deep neural networks typically are, substantial effort has been made to reduce number of parameters these networks require without significantly reducing their accuracy. Some of these have taken a similar approach to the MERA network described above, using tensor decompositions of the fully connected layers.

These include the tensor train models of Novikov et al. (2015) and Garipov et al. (2016). Here we have found replacing a fully connected linear layer with a MERA factorization resulted in superior accuracy for a comparable number of parameters.

More directly related to this MERA model are a number of tree tensor network models (Liu et al. (2017), Levine et al. (2017)). As Section 2 explained, tree tensor networks inconsistently capture correlations on the same length scale, this is the reason for the introduction of disentanglers. Tree tensors do not possess these and we expect them to struggle to capture long range correlations as effectively as MERA.

A MERA works through a process of coarse graining or renormalization. There have been a number of other papers looking at the relationship between renormalization and deep learning. Lin et al. (2017) argue that the effectiveness of deep neural networks should be thought of in terms of renormalization and Mehta & Schwab (2014) demonstrate an exact mapping between the variational renormalization group and restricted Boltzmann machines. In this report we have taken a different approach: only the fully connected layers of the network were replaced with MERA layers.

## 6 DISCUSSION

We have shown that replacing the fully connected layers of a deep neural network with layers based upon the multi-scale entanglement renormalization ansatz can generate significant efficiency gains with only small reduction in accuracy. When applied to the CIFAR-10 data we found the fully connected layers can be replaced with MERA layers with 3900 times less parameters with a reduction in the accuracy of less than $1\%$. The model significantly outperformed compact fully connected layers with $70 - 100$ times as many parameters. Moreover, it outperformed a similar replacement of the fully connected layers with tensor trains, both in terms of accuracy for a given compression and compression for a given accuracy.

An added advantage — not explored here — is that a factorized layer can potentially handle much larger input data sets, thus enabling entirely new types of computation. Correlations across these large inputs can be handled much more efficiently by MERA than by tensor trains. Moreover, a compressed network may provide a convenient way to avoid over-fitting of large data sets. The compression achieved by networks with these factorized layers comes at a cost. They can take longer to train than networks containing the large fully connected layers due to the number of tensor contractions required to apply the factorized layer.

Our results suggest several immediate directions for future inquiry. Firstly, there are some questions about how to improve the existing model. For example, before the MERA layer is used the input is reshaped into a rank-12 tensor. There isn't a well defined method for how to perform this reshaping optimally and some experimentation is necessary. The best way to initialize the MERA layers is also still an open question.

The results presented here are a promising first step for using MERA in a more fundamental way. Since MERA can be viewed as a coarse graining procedure (as explained in Section 2), and image data is often well represented in a hierarchical manner, one possibility would be to simply train a two-dimensional MERA directly on an image dataset, with no reference to a neural network. In Stoudenmire & Schwab (2016) a similar idea was explored with matrix product states being trained directly on MNIST. An alternative possibility would be the replacement of just the convolutional layers of the network with a two-dimensional MERA. Both of these approaches would be closer in spirit to the fundamental ideas about the relationships between quantum physics and machine learning proposed in Lin et al. (2017) and Mehta & Schwab (2014).

Additionally, there has been some work using entanglement measures to explore how correlations are distributed in deep neural networks, and then utilizing these in order to optimize the design of

networks (Liu et al. (2017), Levine et al. (2017)). It would be intriguing to explore such ideas using MERA, for example by using the concrete MERA model explored in this paper, or one of the more ambitious possibilities mentioned above.

We end by noting two facts: any variational approximation to a quantum wavefunction can be used to construct a replacement for linear layers of a network. There are many examples and each may have its sphere of useful application. Moreover, quantum computers of the type being developed currently by several groups are precisely described by a type of tensor network (a finite-depth circuit - and one that may very soon be too large to manipulate classically) and could be used as direct replacement for linear layers in a hybrid quantum/classical neural computation scheme.

ACKNOWLEDGMENTS

This work was supported by the Engineering and Physical Sciences Research Council [grant number EP/P510270/1]. The Titan Xp used for this research was donated by the NVIDIA Corporation. We would like to thank Miles Stoudenmire for many enlightening discussions.

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
