# OpenReview forum: "Compact Neural Networks based on the Multiscale Entanglement Renormalization Ansatz"
_ICLR.cc/2018/Conference — Reject_

### Official Review · AnonReviewer3 · 2017-11-26
**Interesting method but the paper needs to be rewritten**

**Rating:** 5
**Confidence:** 3

**Review:**

In the paper the authors suggest to use MERA tensorization technique for compressing neural networks. MERA itseld in a known framework in QM but not in ML. Although the idea seems to be fruitful and interesting I find the paper quite unclear. The most important part is section 2 which presents the methodology used. However there no equations or formal descriptions of what is MERA and how it works. Only figures which are difficult to understand. It is almost impossible to reproduce the results based on such iformal description of tensorization method. The authors should be more careful and provide more details when describing the algorithm. There was enough room for making the algorithm more clear. This is my main point for critisism.

Another issue is related with practical usefulness. MERA allows to get better compression than TT keeping the same accuracy. But the authors do compress only one layer. In this case the total compression of DNN is almost tha same so why do we need yet another tensorization technique? I think the authors should try tenzorizing several layers and explore whether they can do any better than TT compression. Currently I would say the results are comparable but not definitely better.

UPDATE: The revised version seems to be a bit more clear. Now the reader unfamiliar with MERA (with some efforts) can understand how the methods works. Although my second concern remains. Up to now it looks just yet another tensorization method with only slight advantages over TT framework. Tensorization of conv.layers could improve the paper a lot. I increased the score to 5 for making the presentation of MERA more readable.

---

> ### Author Response · Authors · 2018-01-06
> **Response to AnonReviewer3**
>
>
> Thank you for your helpful comments.
>
> We have adapted the manuscript to include a more comprehensive description of those principles that may not be familiar to a machine learning audience and a more formal description of the MERA layers. We hope that you find the revised version to be more clear.
>
> In the MERA and MPO experiments we compress the two penultimate layers of the network. We have amended the paper to make this more clear.

---

### Official Review · AnonReviewer2 · 2017-11-26
**Interesting idea but more experimental validation required**

**Rating:** 5
**Confidence:** 4

**Review:**

The paper presents a new parameterization of linear maps for use in neural networks, based on the Multiscale Entanglement Renormalization Ansatz (MERA). The basic idea is to use a hierarchical factorization of the linear map, that greatly reduces the number of parameters while still allowing for relatively complex interactions between variables to be modelled. A limited number of experiments on CIFAR10 suggests that the method may work a bit better than related factorizations.

The paper contains interesting new ideas and is generally well written. However, a few things are not fully explained, and the experiments are too limited to be convincing.


Exposition
On a first reading, it is initially unclear why we are talking about higher order tensors at all. Usually, fully connected layers are written as matrix-vector multiplications. It is only on the bottom of page 3 that it is explained that we will reshape the input to a rank-k (k=12) tensor before applying the MERA factored map. It would be helpful to state this sooner. It would also be nice to state that (in the absense of any factorization of the weight tensor) a linear contraction of such a high-rank tensor is no less general than a matrix-vector multiplication.

Most ML researchers will not know Haar measure. It would be more reader friendly to say something like "uniform distribution over orthogonal matrices (i.e. Haar measure)" or something like that. Explaining how to sample orthogonal matrices / tensors (e.g. by SVD) would be helpful as well.

The article does not explain what "disentanglers" are. It is very important to explain this, because it will not be generally known by the machine learning audience, and is the main thing that distinguishes this work form earlier tree-based factorizations.

On page 5 it is explained that the computational complexity of the proposed method is N^{log_2 D}. For D=2, this is better than a fully connected layer. Although this theoretical speedup may not currently have been realized, it perhaps could be achieved by a custom GPU kernel. It would be nice to highlight this potential benefit in the introduction.


Theoretical motivation
Although I find the theoretical motivation for the method somewhat compelling, some questions remain that the authors may want to address. For one thing, the paper talks about exploiting "hierarchical / multiscale structure", but this does not refer to the spatial multi-scale structure that is naturally present in images. Instead, the dimensions of a hidden activation vector are arbitrarily ordered, partitioned into pairs, and reshaped into a (2, 2, ..., 2) shape tensor. The pairing of dimensions determines the kinds of interactions the MERA layer can express. Although the earlier layers could learn to produce a representation that can be effectively analyzed by the MERA layer, one is left to wonder if the method could be made to exploit the spatial multi-scale structure that we know is actually present in image data.

Another point is that although from a classical statistics perspective it would seem that reducing the number of parameters should be generally beneficial, it has been observed many times that in deep learning, highly overparameterized models are easier to optimize and do not necessarily overfit. Thus at this point it is not clear whether starting with a highly constrained parameterization would allow us to obtain state of the art accuracy levels, or whether it is better to start with an overparameterized model and gradually constrain it or perform a post-training compression step.


Experiments
In the introduction it is claimed that the method of Liu et al. cannot capture correlations on different length scales because it lacks disentanglers. Although this may be theoretically correct, the paper does not experimentally verify that the proposed factorization with disentanglers outperforms a similar approach without disentanglers. In my opinion this is a critical omission, because the addition of disentanglers seems to be the main or perhaps only difference to previous work.

The experiments show that MERA can drastically reduce the number of parameters of fully connected layers with only a modest drop in accuracy, for a particular ConvNet trained on CIFAR10. Unfortunately this ConvNet is far from state of the art, so it is not clear if the method would also work for better architectures. Furthermore, training deep nets can be tricky, and so the poor performance makes it impossible to tell if the baseline is (unintentionally) crippled.

Comparing MERA-2 to TT-3 or MERA-3 to TT-5 (which have an approximately equal number of parameters), the difference in accuracy appears to be less than 1 percentage point. Since only a handful of specific MERA / TT architectures were compared on a single dataset, it is not at all clear that we can expect MERA to outperform TT in many situations. In fact, it is not even clear that the small difference observed is stable under random retraining.


Summary
An interesting paper with novel theoretical ideas, but insufficient experimental validation. Some expository issues need to be fixed.

---

> ### Author Response · Authors · 2018-01-06
> **Response to AnonReviewer2**
>
>
> Thank you for your considered response.
>
> We have adapted the manuscript to give a more comprehensive description of the architecture and those principles that may not be familiar to a machine learning audience, including tensor notation, disentanglers and sampling random orthogonal matrices/tensors. We hope that this is now more clear.
>
> The role of the disentanglers, in terms of performance, has not been directly examined. One difficulty is that removing the disentanglers also reduces the number of model parameters thus biasing the comparison. We do not believe this would be a fair comparison. In future work we are planning to more thoroughly examine the role of disentanglers in various architectures.
>
> Whether compression is best achieved by factorizing the weight tensors or constraining or distilling larger models during or after training is an interesting question and we don’t make this comparison. However, using factorization initially would seem to allow for models with more capacity using the same number of parameters and the two approaches are not always mutually exclusive
>
> Regarding your comment about the reshaping of the activation vector from the final convolutional layer. We agree that this is a somewhat arbitrary choice that is also apparent in other tensorization methods. This issue could be avoided by constructing the entire network from tensor components, which we plan to examine in future work.
>
> To compare the MERA and TT factorization methods we used a very simple architecture and basic data augmentation to as best possible isolate the effects of factorization from other design choices. It would indeed be very interesting to test these methods in a more complicated model.
>
> Thank you again for a very helpful and detailed response.

---

### Official Review · AnonReviewer1 · 2017-11-28
**Not very rigorously written, results are mediocre**

**Rating:** 4
**Confidence:** 4

**Review:**

The authors study compressing feed forward layers using low rank tensor decompositions. For instance a feed forward layer of 4096 x 4096 would first be reshaped into a rank-12 tensor with each index having dimension 2, and then a tensor decomposition would be applied to reduce the number of parameters.

Previous work used tensor trains which decompose the tensor as a chain. Here the authors explore a tree like decomposition. The authors only describe their model using pictures and do not provide any rigorous description of how their decomposition works.

The results are mediocre. While the author's approach does seem to reduce the feed forward net parameters by 30% compared to the tensor train decomposition for similar accuracy, the total number of parameters for both MERA (authors' approach) and Tensor Train is similar since in this regime the CNN parameters dominate (and the authors' approach does not work to compress those).

---

> ### Author Response · Authors · 2018-01-06
> **Response to AnonReviewer1**
>
> Thank you for your helpful comments.
>
> We have revised the manuscript to include a more comprehensive description of the MERA decomposition.  We hope that you find this sufficient.
>
> We have now considered the regime in which the convolutional parameters make up a relatively small number of the total number of parameters in the network. In this ablated network the MERA layer also outperforms the tensor train network.

---

### Decision · Program_Chairs · 2018-01-29
**ICLR 2018 Conference Acceptance Decision**

**Decision:**

Reject

**Comment:**

This paper proposes a tree-structured tensor factorisation method for parameter reduction. The reviewers felt the paper was somewhat interesting, but agreed that more detail was needed in the method description, and that the experiments were on the whole uninformative. This seems like a promising research direction which needs more empirical work, but is not ready for publication as is.